# The Failure Mechanisms of Precast Geopolymer after Water Immersion

**DOI:** 10.3390/ma14185299

**Published:** 2021-09-14

**Authors:** Shunfeng Wang, Long Yu, Linglin Xu, Kai Wu, Zhenghong Yang

**Affiliations:** 1School of Materials Science and Engineering, Tongji University, 4800 Cao’an Road, Shanghai 201804, China; 1910617@tongji.edu.cn (S.W.); xulinglinok@hotmail.com (L.X.); wukai@tongji.edu.cn (K.W.); tjzhy92037@163.com (Z.Y.); 2Key Laboratory of Advanced Civil Engineering Materials, Tongji University, Ministry of Education, 4800 Cao’an Road, Shanghai 201804, China

**Keywords:** steel slag, geopolymer, water immersion, molding pressure technology

## Abstract

Precast geopolymers with lower water/binder (0.14), which mainly consists of alkali solution, fly ash (FA) and steel slag (SS), were manufactured through molding pressing technology. The failure mechanisms of precast geopolymers after water immersion were studied by testing the loss of compressive strength, the pH of the leaching solution, the concentration of ions (Na^+^, Ca^2+^, Si^4+^ and Al^3+^), the evolution of phases, pore structure and morphology, and further discussion of the regulation evolution was performed. The results show that the harmful pores (>50 nm) of geopolymers progressively decrease from 70% to 50% after 28 days of water immersion when the content of steel slag increases from 0 to 80 wt.%. Compressive strength of geopolymers sharply reduces in the first 3 days and then increases during the water immersion process, but the phase composition varies slightly. Furthermore, increasing the content of steel slag could decrease the total porosity and further prevent the water resistance.

## 1. Introduction

Geopolymer is a new type of green building material; it can be prepared through mixing aluminosilicate materials with alkaline and acid solution [1,2,3,4]. It is an inorganic polymeric material due to the three-dimensional network structure and amorphous gel (N-A-S-H and C-A-S-H) through the oxygen bridge connection of Si-O-Al-AO. More importantly, geopolymer has excellent mechanical properties and durability, such as high compressive strength, high temperature resistance, low CO_2_ emission and adsorbents [5,6]. It is not only applied in engineering, but also has attracted increasing attention of researchers in recent decades.

Geopolymer also can use solid waste (fly ash, steel slag, blast furnace slag, waste glass power, etc.) as raw materials [7]. Steel slag is a byproduct of the metallurgical industry. China, produces more than 100 million tons of steel slag every year. However, the total capacity factor is below 30% [8,9,10,11]. Large amounts of steel slag accumulation will cause some serious environment problems such as water pollution, occupation of a huge amount of farmland, and waste of resources [12,13]. Thus, it has become a key point of the common concerned problems in the resource utilization and environment protection with the further development of the metallurgy industry. At present, most of the steel slag is used as soil modifiers and subgrade filling materials, and it is also used as raw materials for geopolymer due to the high content of SiO_2_ and Al_2_O_3_ [14,15,16]. Geopolymer also shows excellent mechanical property and durability compared to the ordinary cementitious materials; it also has a great potential in precast building material [17,18,19]. Increasing the curing temperature makes it possible to obtain a higher compressive strength, which can be attributed to the improved dissolution and polycondensation rates. However, conventional curing methods (high temperature and oven curing) could not satisfy higher mechanical property and rapid construction requirements [20,21,22]. Comparing to ordinary concrete, precast building material has several advantages in quality, fast assembling, safety and reduction in manpower demands and economic benefits [23,24].

A higher compressive strength can be realized by molding pressure technology with a lower water/binder ratio. Several researchers have reported that the precast geopolymers can be successfully manufactured by molding pressure technology [23,24,25]. Our previous work found that inorganic-organic polymer composites (IOPC) by mold pressing had lower porosity and higher compressive strength [25]. Furthermore, mold pressing technology could significantly decrease the total porosity and total percentage of harmful pores (50–200 nm) and (>200 nm) of geopolymers. Zhang et al. studied the pore structure of hardened cement paste and the microstructure regulation mechanism and defect reduction after pressure compaction [26]. Results indicated that the pore surface fractal dimension of cement paste become multisegmental fractal and scale dependent after being pressurized as compact molding, and the irregular drift behavior in the transition region was not obvious. The compressive strength of geopolymer is 134 MPa by using simultaneous heating and pressing techniques [27,28,29]. In this paper, mold pressure technology to manufacture geopolymers is evaluated; it can not only reduce energy consumption and obtain higher compressive strength, but also shorten the curing time. However, the low w/b in mold pressing also results in other problems, such as cracking, compressive strength loss, etc. Although mold pressing technology has promising applications in precast geopolymers, the water resisting property of so prepared geopolymer was studied insufficiently. The report given by Silva et al. [30] discussed the water-resistance of alkali-activated materials obtained from tungsten mine waste mud; it found a significant decrease of compressive strength-a 70% loss after 7 days of water immersion. The water-resistance not only hindered polycondensation kinetics, but also led to molecular destabilization of the geopolymer matrix.

Considering the previous discussion, it is essential to explore the failure mechanism of precast geopolymer via molding pressure technology after water immersion. The molding pressure was varied to obtain different ultimately compressive strengths of the geopolymer. We further studied the effect of pressure on compressive strength, pH, dissolution properties of precursor blends, crystal phase, pore structure and morphology of precast geopolymer before and after water immersion. Exploring the failure mechanism of precast geopolymer offers a new path to develop a high performance and water-resistant geopolymer composites.

## 2. Experimental Procedure

### 2.1. Materials

Class F fly ash (FA) used in this study was obtained from Baotian new building materials Co. Ltd. (Shanghai, China). Steel slag (SS) was supplied by Jinan steel group Co. Ltd. (Shanghai, China). As illustrated in Figure 1, the main phase compositions of FA were quartz, mullite and hematite, while SS has only an amorphous composition. The main oxide compositions of fly ash and steel slag were determined by X-ray fluorescence (XRF), and the results are shown in Table 1. The total content of SiO_2_ and Al_2_O_3_ of FA is 79.93%. The main oxide compositions of SS are SiO_2_ and Fe_2_O_3_. Figure 2 shows the morphology of the FA and SS. FA contains more regular spherical particles and few irregular particles. It can be seen that SS consists mainly of lumpy particles.

In this study, NaOH solution and sodium silicate solution were mixed to prepare the alkali solution. The NaOH solution was made by mixing analytical grade sodium hydroxide (≥98% purity) in tap water to a concentration of 8 mol/L. The sodium silicate solution was obtained from PQ Australia Pty. Ltd. (Southern Queensland, Australia), with an original modulus of 3.2. The main compositions of sodium silicate solution are 8.5 wt.% SiO_2_, 26.5 wt.% Na_2_O and 65 wt.% H_2_O.

### 2.2. Synthesis of Geopolymers

The proportions of FA, SS, NaOH solution and Na_2_SiO_3_ and water/binder are tabulated in Table 2. The ratio of the water (including the mass of NaOH solution and water in water glass) and binder (including the mass of fly ash and steel slag) is 0.14. The water includes the extra water and the water in water glass. For all geopolymers, the mixture of FA and SS was mixed for 3 min to obtain homogeneous powders before adding the alkali activation solution. Then, the homogeneous powders and alkali activation solution were mixed for 2 min at a high speed to prepare homogeneous pastes. The fresh paste was poured into a cubic stainless steel mold (20 × 20 × 20 mm^3^), which was exposed to different uniaxial pressures (40, 60 and 80 kN, respectively) for 2 min using the loading rate of 500 N/s. As we all know, the mold pressing technique was widely used in the preparation of ceramics, but only minimally used in cement production. This technology can decrease the total porosity and the water/binder ratio of samples. All geopolymers, after demolding, were cured at 60 °C for 1 day, and then cured at room temperature with a relative humidity 95% for 3 days.

### 2.3. Testing and Characterization

To qualify the water resisting property, samples are mixed with deionized water (100 mL). Six samples were immersed in deionized water for 3, 7, 14 and 28 days. Then, the geopolymers were taken out at different periods of water immersion, compressed and compressive strength tests were performed on the cubes using a universal testing machine with a loading rate of 500 N/s. The pH of the leaching solution at different periods of 3, 7, 14 and 28 days was also measured using an intelligent pH meter. The diluted solution gave suitable Na^+^, Ca^2+^, Si^4+^ and Al^3+^ concentrations for inductively coupled plasma (ThermofiSher scientific, Massachusetts, USA).

The cumulative pore volumes and pore size distribution of the specimens were determined by Micromeritics Instrument Corp (Quantachrome, VA, USA). First, the samples were crushed into pieces between 3 to 5 mm. Then, they were cooled in liquid nitrogen for 10 min, and immediately transferred to the vacuum dryer. The pressure range of the equipment could be set to 0.14–227.54 MPa, the surface tension of the mercury was 0.48 N/m, and the contact angle between the mercury and pore wall was 140°.

X-ray diffraction (XRD) (Rigaku International Corporation, Tokyo, Japan) was conducted to identify the crystalline phases of geopolymers after water immersion. The powder samples were affixed to a slide and placed in a D/max 2550VB3+/PC X-ray diffractometer with a Cu-Kα radiation operating at 40 kV and 100 mA. The samples were scanned in the region from 5 to 70° with a step of 0.02° at a rate of 2°/min. A field emission scanning electron microscope (FE-SEM, JEOL JSE-7500F) measures the microstructures of specimens.

## 3. Results and Discussion

### 3.1. Choosing Appropriate Molding Pressure

The effect of molding pressure on the compressive strength of selected FA60SS40 at different curing ages is shown in Figure 3. It is noteworthy that as the molding pressure increases, the compressive strength gradually increases. Compressive strength of precast geopolymers only increases by a small amount with curing age. If the molding pressure is above 80 kN in the experiment process, the alkali solution will leak out due to the higher pressure [25]. Therefore, the molding pressure of 80 kN was ultimately chosen.

### 3.2. Pore Structure Analysis

Figure 4 presents the measured pore size distributions of precast geopolymers at the age of 3 days and after different periods of water immersion. Figure 4a clearly shows that the peak value in the differential pore volume curve corresponds to the most probable pore diameter of precast geopolymers, and gradually shifts to a higher diameter as the content of steel slag increases at the age of 3 days. The most probable pore diameter of precast geopolymers at 3 days is 1335, 1392, 1815 and 1796 nm with the content of steel slag increased from 0 to 80 wt.%, respectively. Meanwhile, not only the peak value sharply decreases but also the total porosity of precast geopolymers decreases while the content of steel slag is above 20 wt.%. Therefore, the compressive strength of precast geopolymers increases slightly with the addition of the steel slag from 0 to 80 wt.% (Figure 4) while the most probable pore diameter increases progressively. The main reason is that spherical fly ash particles will increase the total porosity, while adding steel slag increases the compaction and decreases the total porosity of geopolymers [31,32,33]. After water immersion (Figure 4b–e), the peak value of precast geopolymers changes little when fly ash is replaced by 20 wt.% steel slag. However, the total porosity of precast geopolymers gradually increases, which results in a decrease in the compressive strength. Furthermore, the most probable pore diameter of precast geopolymers remarkably shifts to a fine diameter. It is noteworthy that the most probable pore diameter of FA100 at 3 days and after different periods of water-resistance is 1335, 1191, 1153, 1171, and 1251 nm, respectively. The water immersion could refine the pore sizes due to the geopolymerization process continuing progressively. However, some large pores are also present in this system; the pore sizes can be reduced by high-temperature treatments of geopolymers and hydrothermal transformation into dense ceramic phases.

In addition, a careful examination of our study and the current literature indicates that the pore size could be divided into four principal zones according to Wu et al. [34]: harmless pores <20 nm, few harmful pores 20–50 nm, harmful pores 50–200 nm and more harmful pores >200 nm. Figure 5 displays the total porosity and pore size percentage of precast geopolymers at 3 days and at different periods of water immersion. The total porosity of precast geopolymers increases after 3 days of water immersion except for FA100. Increasing the period of water immersion, the total porosity of precast geopolymers will gradually decrease because water increases the contact points of reaction between raw materials and alkali metal ions.

For comparison, the harmless pores and few harmful pores of geopolymers sharply increases with the period of water immersion. Furthermore, the compressive strength of all geopolymers will increase after water immersion. The reason is that N–A–S–H and C–S–H gels could fill in the pore to refine the pore diameter as the alkali ions continue to react with the soluble silicon and aluminum so that the percentage of more harmful pores observably decreases from 70% to 50%. Moreover, the percentage of the more harmful pores progressively decreases after 28 days water immersion when the content of steel slag increases from 0 to 80 wt.%.

### 3.3. Loss of Compressive Strength after Water Immersion

Compressive strength is a key physical quantity for evaluating the water resisting property of cement-based materials. The compressive strengths of precast geopolymers at the age of 3 days and after different periods of water immersion are shown in Figure 6. As can be seen from Figure 6, the compressive strength of precast geopolymers gradually increases with the increase of steel slag at 3 days. The compressive strength of FA20SS80 reaches the highest value (56.78 MPa). The rising in the compressive strength of precast geopolymers is the result of the total porosity and the percentage of harm pore of precast geopolymers decreasing little by little, which will be discussed in Section 3.2. However, the compressive strength of precast geopolymers after water immersion for 3 days decreases dramatically, which indicates that the internal deterioration of precast geopolymers is more severe than that of 3 days. The results indicate that precast geopolymers absorb a large amount of water through capillary pores because the lower ratio of water/binder causes the difference in humidity, which will release a large amount of OH^−^, Na^+^, Si^4+^ and Al^3+^ and break the bridges between the geopolymerization products and weaken the gel structure. As presented in Figure 6, the compressive strength of precast geopolymers gradually increases with increasing of the period of water immersion. This is because water increases the contact points of reaction between raw materials and alkali metal ions, which determines the polycondensation kinetics. The compressive strength of precast geopolymers after 28 days of water immersion is 19.2, 21.5, 26.31, 29.8 and 30.9 MPa with increasing content of steel slag from 0 to 80 wt.%, respectively. Meanwhile, it can be concluded that adding steel slag is beneficial to prevent the loss of compressive strength caused by water immersion. A study involving calcined kaolin indicated that nonevaporable water was necessary to keep the strength stable and the optimum content was about 7.4% [35].

Summarily, the compressive strength of precast geopolymers in water is sharply increased at the age of 3 days. However, the percentage of the fall was equal; geopolymer with higher SS dosage only had a higher value of the reference compressive strength. Then it will gradually increase with the period of water immersion. Additionally, adding steel slag has a positive influence on the compressive strength of precast geopolymers at the age of 3 days and at different periods of water immersion.

### 3.4. Evaluation of the Water-Resistance

#### 3.4.1. The pH Values of Leaching Solutions

Figure 7 presents the pH values of the leaching solution. In general, the pH values of the leaching solution are higher while the specimens were immersed in water for 3 days. The total OH^−^ content in the first 3 days has already reached above 80% of that of 28 days. The key reason is that a large amount of OH^−^ is released on the surface and inside of specimens, which results in a higher pH in the first 3 days. With increasing of the period of water immersion, the pH values slowly increase [36]. The pH values of geopolymers from 3 days to 28 days after water immersion increase by 0.21, 0.20, 0.32, 0.34 and 0.48 respectively with increasing content of steel slag from 0 to 80 wt.%. However, the pH value of FA20SS80 after 28 days of water immersion is below that of FA100 after 3 days. Hence, all precast geopolymers made with steel slag exhibit a lower leachate pH after 3 days of water immersion, which means that adding steel slag not only increases the compact degrees of geopolymers, but also provides some benefits in decreasing the leaching rate of OH^-^ due to the lower porosity.

#### 3.4.2. The pH Values of Leaching Solutions

The ions content of Na^+^, Ca^2+^, Si^4+^ and Al^3+^ leached from the precast geopolymers at different periods of water immersion are presented in Figure 8. Figure 8a shows the evolution of Na^+^ content, where it can be observed that the content of Na^+^ gradually increases with an increasing period of water immersion [26]. At the same time, the content of Na^+^ increases firstly and then decreases with increasing steel slag dosage. The soluble Na^+^ content of geopolymers after 3 days of leaching accounts for more than 80 wt.% of that of 28 days. Furthermore, the dissolution of a large number of Na^+^ in the leachate will destroy the gel structure of geopolymers. The Ca^2+^ content of precast geopolymers is shown in Figure 8b. It can be observed that the soluble Ca^2+^ of precast geopolymers increases slightly during the water immersion process. The total Ca^2+^ content is below 6.5 × 10^−3^ g although steel slag contains a large amount of Ca^2+^. After a 28-day water immersion, the content of Ca^2+^ sharply decreases when the content of steel slag is above 40 wt.%. The reason is that increasing the content of steel slag will increase the most probable pore diameter of precast geopolymers, and the large amount of Ca^2+^ will react with soluble Si to form C-S-H in the surface of specimens. The Si^4+^ content of precast geopolymers observed in Figure 8c is progressively increased with increasing period, while it will increase firstly and then decrease with adding the steel slag. It is noteworthy that the total content of Si^4+^ increases as the content of steel slag increases. This indicates that steel slag contains amounts of soluble Si^4+^ comparing to fly ash. At the same time, the soluble Si^4+^ rate gradually decreases when the period of water immersion exceeds 14 days, which indicates that the soluble silicate has limited influence on the leaching of Ca^2+^ and Al^3+^. However, the Al^3+^ amount in the leaching solution is quite low, and gradually decreases with increasing steel slag content due to the low Al^3+^ content in steel slag and the dilution effect (Figure 8d). Increasing the leaching time, the total Al^3+^ content will gradually increase. The total content of Al^3+^ is below 3 × 10^−3^ g for all precast geopolymers, and the leaching content will increase gradually with increasing water immersion.

Based on these results, from the ions content of the leaching solution shown in Figure 6, the content of Ca^2+^ and Al^3+^ gradually decreases and the growing rate of Si^4+^ gradually slows down while the content of steel slag is above 40 wt.%. It can be concluded that Ca^2+^, Al^3+^ and Si^4+^ are formed C-A-S-H, which means that the compressive strength of geopolymers begin to increase.

#### 3.4.3. XRD Analysis

Figure 9 plots the XRD pattern of FA60SS40 at the age of 3 days and after different periods of water immersion. For all precast geopolymers, the crystalline phases are quartz and mullite, contributed by fly ash. However, there are amorphous N-A-S-H gels presented, as shown in the broad hump in the range of 20 to 40° 2 degree [37]. At the same time, there is no significant difference among the diffraction patterns of FA60SS40 after different periods of water immersion comparing to 3 days, indicating that the influence of water immersion on the phase compositions and crystallinity of geopolymers is negligible.

#### 3.4.4. Morphology of Precast Geopolymers after Water Immersion

The morphology of precast geopolymers at the age of 3 days and after water immersion is shown in Figure 10. In Figure 10a, there are some pores and unreacted spherical fly ash particles on the surface of FA100 at the age of 3 days. Furthermore, the surface of particles is covered by N-A-S-H gels. Increasing the period of water immersion, the leaching of residual soluble silicon aluminum will not only put off polycondensation kinetics, but also lead to molecular destabilization of the precast geopolymer matrix. Specimens FA60SS40 and FA20SS80 show a more compact microstructure than that observed in FA100, as shown in Figure 10a,d,g. Considering the pore structure analysis and morphology of precast geopolymers, it can be found that adding the steel slag is beneficial to refine pore diameters and improves the compressive strength. Based on the observation of Figure 10d–f, the morphology of FA60SS40 changes a little at the age of 3 days and after water immersion. Moreover, steel slag and fly ash are uniformly piled up. It is also worth noting that the fly ash particles of FA20SS80 after 14 days water immersion are still covered by gel phases, as shown in Figure 10h. Thus, adding steel slag could improve the formation of (C, N)-A-SH-, prevent water attack and decrease the total porosity of precast geopolymers.

## 4. Conclusions

Precast geopolymers with a high early compressive strength which was produced via molding pressure technology have been fabricated in this work. The failure mechanisms of precast geopolymers after water immersion was systematically investigated via compressive strength, pH, ions dissolution, phase compositions, pore structure and morphology. The following conclusions can be drawn:

(1) The compressive strength of geopolymers gradually increase with the molding pressure and the content of steel slag. The ultimate molding pressure of 80 kN was selected in this work, and the optimal compressive strength of FA20SS80 is obtained after curing 3 days, which is 56.78 MPa.

(2) Adding the steel slag could prevent the loss of compressive strength after water immersion, 28 days compressive strength of geopolymers after water immersion could retain 50% of the reference.

(3) With extending of the period of water immersion, the pH of leaching solution gradually decreases with the increase of steel slag replacement level, but the phase composites only change slightly. The compressive strength of precast geopolymers increase with the period of water immersion and increasing content of steel slag.

(4) After 28 days of water immersion, the percentage of the more harmful pores (>50 nm) of precast geopolymers progressively decreases from 70 to 50% with the replacement of fly ash by steel slag from 0 to 80 wt.%. Furthermore, the percentage of the more harmful pores gradually decreases as the content of steel slag increased.

## Figures and Tables

**Figure 1 materials-14-05299-f001:**
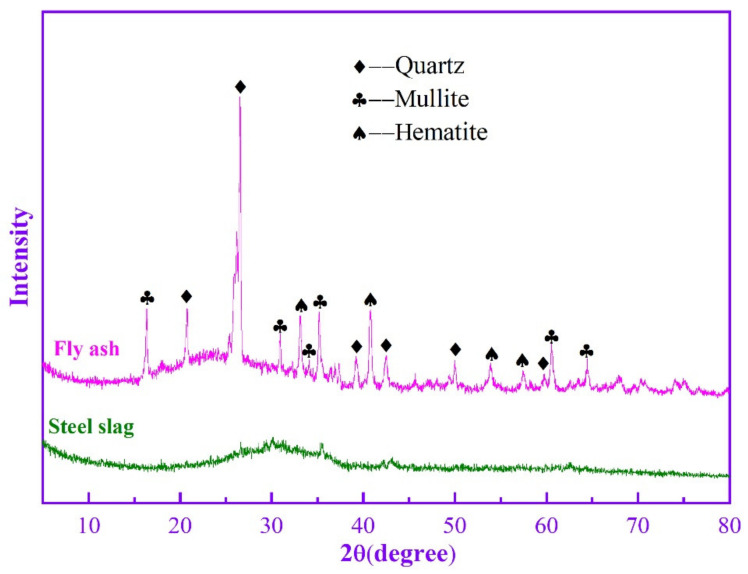
XRD patterns of FA and SS.

**Figure 2 materials-14-05299-f002:**
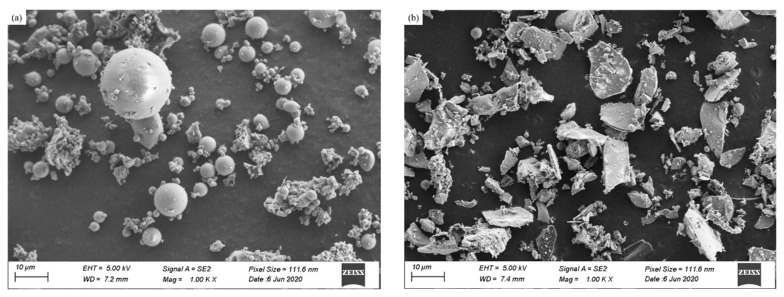
Particle morphologies as determined by secondary electron imaging in SEM: (**a**) FA and (**b**) SS.

**Figure 3 materials-14-05299-f003:**
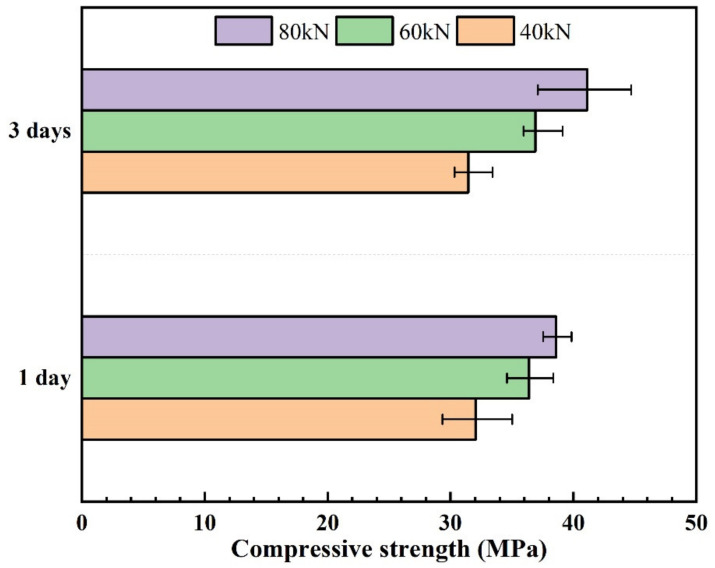
Compressive strength of FA60SS40 at different molding pressures.

**Figure 4 materials-14-05299-f004:**
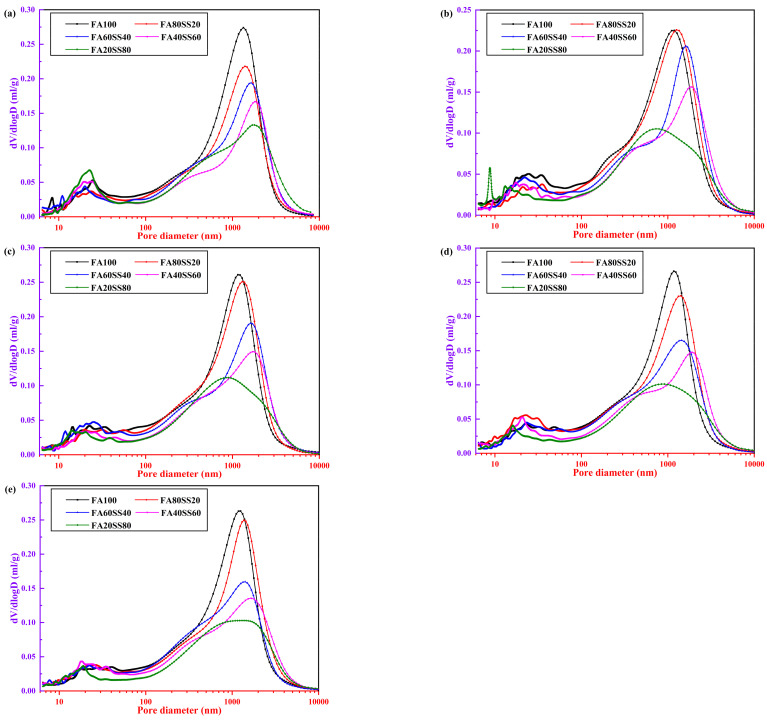
Pore size distributions of precast geopolymers (**a**) at the age of 3 days and at different periods of water-resistance (**b**) 3 days, (**c**) 7 days, (**d**) 14 days and (**e**) 28 days.

**Figure 5 materials-14-05299-f005:**
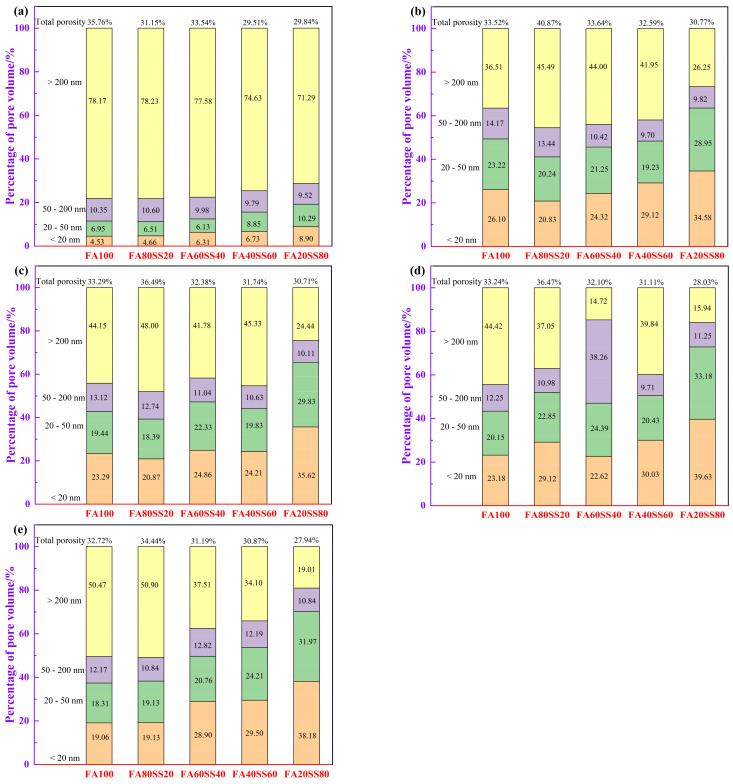
Total porosity and pore size percentage of precast geopolymers (**a**) at the age of 3 days and at different periods of water immersion: (**b**) 3 days, (**c**) 7 days, (**d**) 14 days and (**e**) 28 days.

**Figure 6 materials-14-05299-f006:**
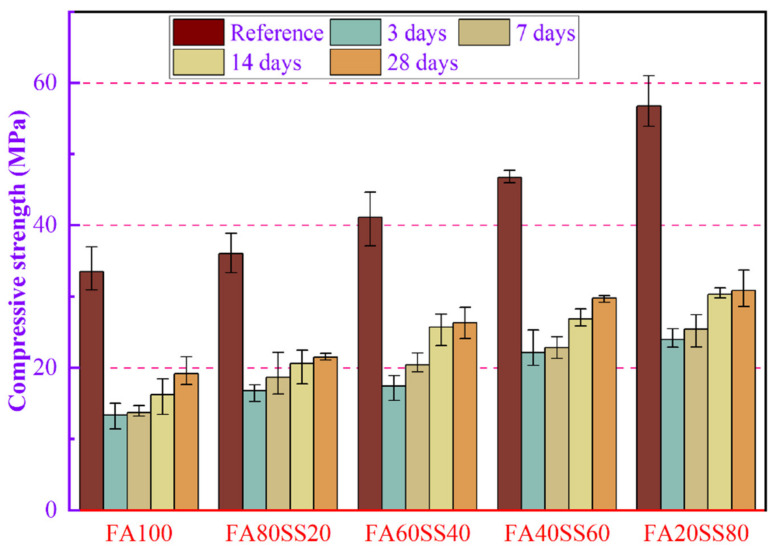
Compressive strength of precast geopolymers at 3 days and after different periods of water-resistance.

**Figure 7 materials-14-05299-f007:**
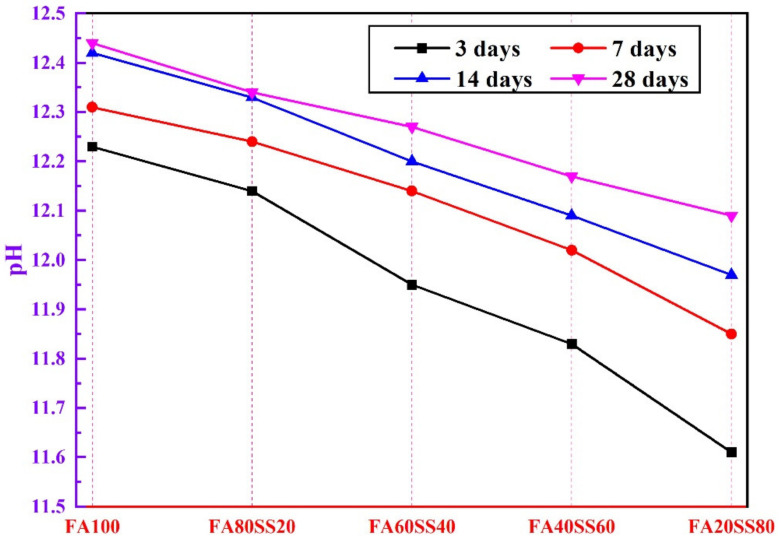
The pH values of leaching solutions of geopolymer.

**Figure 8 materials-14-05299-f008:**
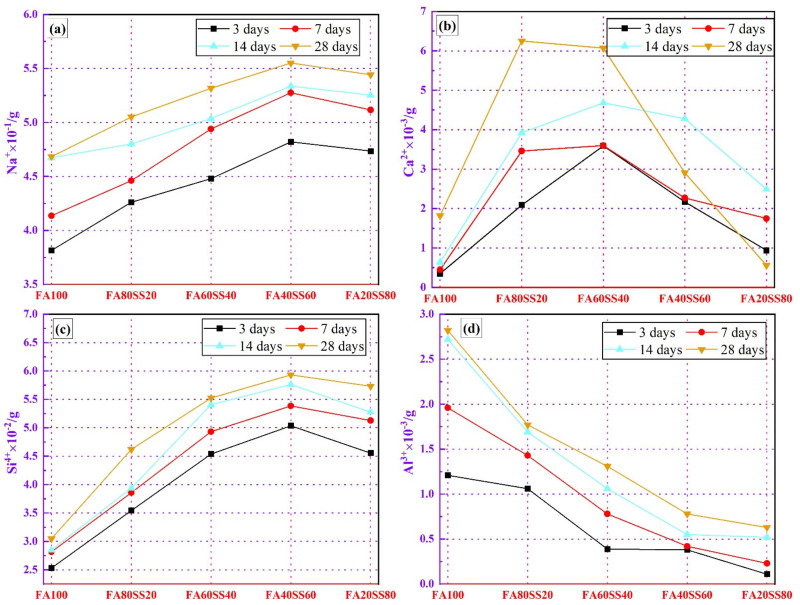
The content of (**a**) Na^+^, (**b**) Ca^2+^, (**c**) Si^4+^ and (**d**) Al^3+^ leached from the precast geopolymers at different periods of water-resistance.

**Figure 9 materials-14-05299-f009:**
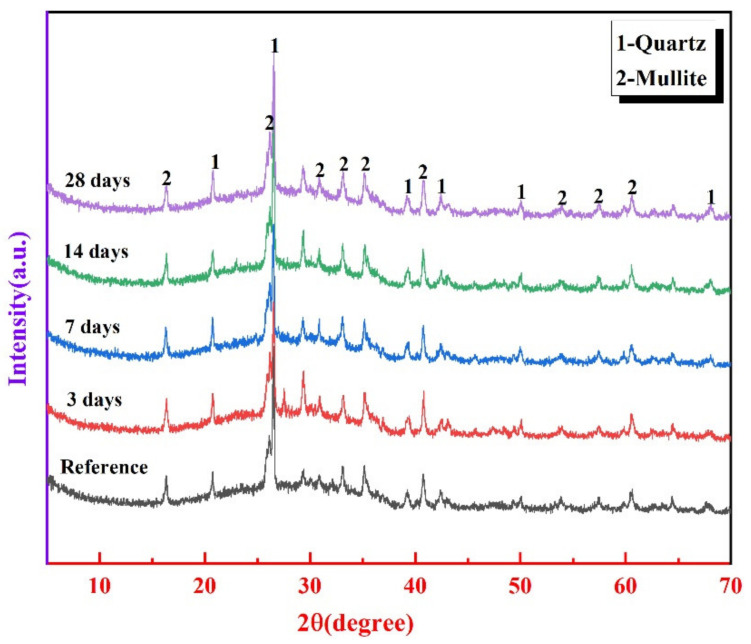
XRD patterns of FA60SS40 at the age of 3 days and after different period in water immersion.

**Figure 10 materials-14-05299-f010:**
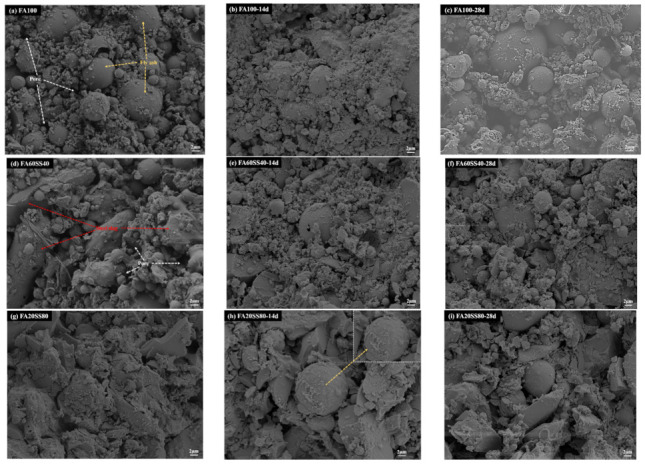
Morphology of geopolymers at the age of 3-day and after water immersion.

**Table 1 materials-14-05299-t001:** Compositions of fly ash and steel slag by XRF analysis.

Oxide/wt.%	SiO_2_	Al_2_O_3_	Fe_2_O_3_	SO_3_	TiO_2_	MgO	P_2_O_5_	Na_2_O	CaO	Other	LOI ^1^
Fly ash	52.50	27.43	3.42	1.47	1.33	1.05	0.47	0.51	7.88	1.62	2.33
Steel slag	33.23	8.06	31.90	1.44	0.48	3.61	0.26	2.41	7.38	8.80	2.43

^1^ Loss on ignition at 1000 °C, wt.%.

**Table 2 materials-14-05299-t002:** Mix proportions of the geopolymer.

Sample ID	Fly Ash/g	Steel Slag/g	NaOH (8 mol/L)/g	Na_2_SiO_3_/g	w/b
FA100	100	0	7	10.64	0.14
FA80SS20	80	20	7	10.64	0.14
FA60SS40	60	40	7	10.64	0.14
FA40SS60	40	60	7	10.64	0.14
FA20SS80	20	80	7	10.64	0.14

## Data Availability

The data presented in this study are available on request from the corresponding author.

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
