# Peer review of "The Failure Mechanisms of Precast Geopolymer after Water Immersion"

_materials, 2021, doi:10.3390/ma14185299_

Round 1

Reviewer 1 Report

This article is focused on hot issue of geopolymer rupture mechanism after water immersion. In the introduction the geopolymers are adequately introduced, and overall current knowledge of this matter is summarized. The reason of the need of presented study is sufficiently rationalized alongside. Used references are topical, their amount is sufficient, and they adequately support presented ideas. The second chapter covers studied raw-materials, used production process, as well as the experimental program. All is described in detail, and overall quality is top. Following chapter 3 with results and discussion is also well-written, and reached results are, in general, properly discussed. The final conclusions are based on the presented results (and except the number 2) are appropriate.

However there are several point, that could be solved before accepting this paper. There are as follows:

  • I have only one minor comment dealing with phase analysis of FA. There were three quite high diffractions which were not determined (about  31, 60 and 65 2θ).
  • I would like to appreciate the “3.1 choosing appropriate molding pressure” chapter, which contributed to the overall high quality of this article.
  • I would like to recommend careful review of the chapter 3.2, some parts don’t make sense: e.g. “the compressive strength of precast geopolymers in water is sharply increased at the age of 3 days”. Moreover, I cannot entirely agree with the conclusion of beneficial effect of SS on compressive strength decrease due to the water immersion, it seems that the percentage of the fall was equal, material with higher SS dosage only had higher value of the reference compressive strength.
  • I also would like to suggested the rearrangement of presented results. Compressive strength as well as the leaching solution characteristics are linked to the pore structure analysis, thus it would be more consistent to first present the porosity and then the connected properties.
  • In the chapter 3.2.2 there is presented that the high amount of Na+ dissolution lead to the compressive strength fall. However more dissolved Na+ shows material FA20SS80 which was here-in-above presented as the least deteriorated. Alike as the time of immersion is prolonged, the higher amount of dissolved Na+ is presented, but the compressive strength is meanwhile increased. This should be clarified as the presented conclusions are not consistent and corresponding to each other.
  • Only minor issue is quality of the SEM image of FA100 after 28 days (Figure 10/c), this picture is of bad brightness.

Nevertheless the quality of the presented article is suitable and thus I recommend only minor revision before accepting.

Author Response

Point 1: I have only one minor comment dealing with phase analysis of FA. There were three quite high diffractions which were not determined (about 31, 60 and 65 2θ).

Response 1: As suggested, I have revised the Figure 1 on page 3.

Point 2: I would like to appreciate the “3.1 choosing appropriate molding pressure” chapter, which contributed to the overall high quality of this article.

Response 2: As suggested, I have replaced “Choosing an appropriate molding pressure” to “choosing appropriate molding pressure” in the revised manuscript.

Point 3: I would like to recommend careful review of the chapter 3.2, some parts don’t make sense: e.g. “the compressive strength of precast geopolymers in water is sharply increased at the age of 3 days”. Moreover, I cannot entirely agree with the conclusion of beneficial effect of SS on compressive strength decrease due to the water immersion, it seems that the percentage of the fall was equal, material with higher SS dosage only had higher value of the reference compressive strength.

Response 3: Thanks for your problem. I have revised the content in chapter 3.2, such as, “Summarily, the compressive strength of precast geopolymers in water is sharply in-creased at the age of 3 days. But, the percentage of the fall was equal, geopolymer with higher SS dosage only had higher value of the reference compressive strength. Then it will gradually increase with the period of water immersion. Besides, adding steel slag has a positive influence on the compressive strength of precast geopolymers at the age of 3 days and at different period of water immersion.”

Point 4: I also would like to suggested the rearrangement of presented results. Compressive strength as well as the leaching solution characteristics are linked to the pore structure analysis, thus it would be more consistent to first present the porosity and then the connected properties.

Response 4: As suggested, I have revised the rearrangement of presented results in the revised manuscript.

Point 5: In the chapter 3.2.2 there is presented that the high amount of Na+ dissolution lead to the compressive strength fall. However more dissolved Na+ shows material FA20SS80 which was here-in-above presented as the least deteriorated. Alike as the time of immersion is prolonged, the higher amount of dissolved Na+ is presented, but the compressive strength is meanwhile increased. This should be clarified as the presented conclusions are not consistent and corresponding to each other.

Response 5: As suggested, I have revised the chapter 3.2.2 in the revised manuscript.

Point 6: Only minor issue is quality of the SEM image of FA100 after 28 days (Figure 10/c), this picture is of bad brightness.

Response 6: As suggested, I have revised the picture in the revised manuscript.

Reviewer 2 Report

In the paper entitled “The failure mechanisms of precast geopolymer after water immersion” by Wang et al., the authors prepared geopolymers consisting of alkali solution, fly ash, and steel slag by molding pressure technology, and further investigated their structure and property evolution during water immersion by adopting different kinds of characterization methods. In general, this work is solid, self-consistent and rich in data, and the results will be meaningful for researchers in the field of geopolymers and waste immobilization. However, before acceptance, the authors should carefully address the following questions, dramatically improve the writing, and correct the numerous grammatical errors.

  1. On page 2, the sentence regarding the size of harmful pores of geopolymers seems contradictory. In addition, is the w/b ratio refers to water to binder ratio? Please define it in the manuscript.
  2. On page 3, the authors mentioned “All geopolymers after demolding were cured at 60 °C for 1 day, and then cured at room temperature with a relative humidity 95% for 3 days.” Please explain what is the purpose of curing geopolymers in the high humidity environment for 3 days?
  3. In section 3.3.1, the authors studied the amount of OH- in the leaching solutions, previous studies have reported that PH values of leaching solutions also impact the compressive strength of geopolymers. Please discuss the potential impact of the PH value of the leaching solution on the compressive strength of geopolymers.
  4. On page 9-10, the authors introduced the definitions of harmless pores (< 20 nm), few harmful pores (20-50 nm), harmful pores (50-200 nm) and more harmful pores (> 200 nm). However, as shown in Figure 8 and Figure 9, all precast geopolymers contain large pores even exceeding 1000 nm. Although the introduction of steel slag is beneficial for reducing the pore sizes, these relatively large pores are still adverse for their mechanical properties and potential applications. The authors should point out possible strategies to further reduce the pore sizes (e.g., by high-temperature treatments of geopolymers and hydrothermal transformation into dense ceramic phases).
  5. On page 9, the wording “mineralogical properties” is too general. Better use “phase compositions and crystallinity”.

Author Response

Point 1: On page 2, the sentence regarding the size of harmful pores of geopolymers seems contradictory. In addition, is the w/b ratio refers to water to binder ratio? Please define it in the manuscript.

Response 1: As suggested, I have replaced “water/binder ratio” to “w/b ratio” in the revised manuscript.

Point 2: On page 3, the authors mentioned “All geopolymers after demolding were cured at 60 °C for 1 day, and then cured at room temperature with a relative humidity 95% for 3 days.” Please explain what is the purpose of curing geopolymers in the high humidity environment for 3 days?

Response 2: Thanks for your problem, I mainly compare the compressive strength with increasing the curing time at room temperature. Furthermore, curing at the room temperature is the most common method. And, it also reduces the cost of precast geopolymers.

Point 3: In section 3.3.1, the authors studied the amount of OH- in the leaching solutions, previous studies have reported that PH values of leaching solutions also impact the compressive strength of geopolymers. Please discuss the potential impact of the PH value of the leaching solution on the compressive strength of geopolymers.

Response 3: Thanks for your problem, the amount of OH- in the leaching solution comes from the inter of sample. Furthermore, the alkali ion is beneficial to the geopolymerization. Large amount of alkali ion leaches will prevent the geopolymerization and the gel phase will decompose in later curing time. Next, I mainly study the failure mechanism of precast geopolymer after water immersion at the long curing time.

Point 4: On page 9-10, the authors introduced the definitions of harmless pores (< 20 nm), few harmful pores (20-50 nm), harmful pores (50-200 nm) and more harmful pores (> 200 nm). However, as shown in Figure 8 and Figure 9, all precast geopolymers contain large pores even exceeding 1000 nm. Although the introduction of steel slag is beneficial for reducing the pore sizes, these relatively large pores are still adverse for their mechanical properties and potential applications. The authors should point out possible strategies to further reduce the pore sizes (e.g., by high-temperature treatments of geopolymers and hydrothermal transformation into dense ceramic phases).

Response 4: As suggested, I have added “However, some large pores also present this system, it can reduce the pore sizes by high-temperature treatments of geopolymers and hydrothermal transformation into dense ceramic phases.” in the revised manuscript.

Point 5: On page 9, the wording “mineralogical properties” is too general. Better use “phase compositions and crystallinity”.

Response 5: As suggested, I have replaced “phase compositions and crystallinity” to “mineralogical properties” on page 9 in the revised manuscript.

Reviewer 3 Report

In this paper, the precast geopolymers which is mainly consist of alkali solution, fly ash (FA) and steel slag (SS) were manufactured through molding pressing technology. Overall speaking, this is a very interesting paper. The experiments are well designed and the conclusions are well supported by experimental results. However, there are some problems which the authors may look into.

  1. The literature should include more recent studies. For example, the use of waste glass power as a raw material to produce geopolymer (alkali-activated materials) has gained significant attention. (e.g., "Alkali-activated slag supplemented with waste glass powder: Laboratory characterization, thermodynamic modelling and sustainability analysis. Journal of Cleaner Production286, p.125554.").
  2. Based on my experience, the water to binder ratio (0.14) is extremely low. How did you blend the geopolymer paste with such a low water content?
  3. The molding pressure method is not well explained in the paper. The authors may describe how the molding pressure was applied, what kind of special molds were used in the section of materials and methods. 
  4.  Why steel slag could prevent the loss of compressive strength after water immersion?

Author Response

Point 1: The literature should include more recent studies. For example, the use of waste glass power as a raw material to produce geopolymer (alkali-activated materials) has gained significant attention. (e.g., "Alkali-activated slag supplemented with waste glass powder: Laboratory characterization, thermodynamic modelling and sustainability analysis. Journal of Cleaner Production, 286, p.125554.").

Response 1: As suggested, I have added the recent study in the revised manuscript.

Point 2: Based on my experience, the water to binder ratio (0.14) is extremely low. How did you blend the geopolymer paste with such a low water content?

Response 2: Thanks for your problem. First, the powder is mixed by high speed mixer. Then, the geopolymer is prepared through molding pressure technology to ensure the wetting powder.

Point 3: The molding pressure method is not well explained in the paper. The authors may describe how the molding pressure was applied, what kind of special molds were used in the section of materials and methods.

Response 3: As suggested, I have added the molding pressure method in the revised manuscript.

Point 4: Why steel slag could prevent the loss of compressive strength after water immersion?

Response 4: Thanks for your problem, the mainly reason is that the crystal phase of steel slag is amorphous, which has higher activity. At the same time, steel slag also can decrease the total porosity of geopolymers.

Round 2

Reviewer 1 Report

In my opinion, author successfully answered to my comments and adequately  improved the publication. On that account I recommend to accept this article.

Reviewer 3 Report

This paper has been revised based on the comments and is good for publication.